# Learning 4D Embodied World Models

## Abstract

In this paper, we present a 4D embodied world model, which takes in an image observation and language instruction as input and predicts a 4D dynamic mesh predicting how the scene will change as the embodied agent performs actions based on the given instructions. In contrast to previously learned world models which typically generate 2D videos, our 4D model provides detailed 3D information on precise configurations and shape of objects in a scene over time. This allows us to effectively learn accurate inverse dynamic models for an embodied agent to execute a policy for interacting with the environment. To construct a dataset to train such 4D world models, we first annotate large-scale existing video robotics dataset using pretrained depth and normal prediction models to construct 3D consistent 4D models of each video. To efficiently learn generative models on this 4D data, we propose to train a video generative model on this annotated dataset, which jointly predicts RGB-DN (RGB, Depth, and Normal) for each video. We then present an algorithm to directly convert generated RGB, Depth and Normal images into high-quality dynamic 4D mesh models of the world. We illustrate how this enables us to predict high-quality meshes consistent across both time and space from embodied scenarios, render novel views for embodied scenes, as well as construct policies that substantially outperform those from prior 2D and 3D models of the world. Our code, model, and dataset will be made publicly available. Video results can be found at our website: https://4d-worldmodel.github.io/.

## 1 Introduction

The ability to simulate and construct learned models of the world (Ha & Schmidhuber, 2018; Yang et al., 2023a; Zheng et al., 2024; Xiang et al., 2024) opens a rich set of opportunities for constructing intelligent embodied agents. Such models enable flexible policy synthesis (Du et al., 2024; Liang et al., 2024), data simulation and generation (Yang et al., 2023a; Zhu et al., 2024), and flexible long-horizon planning (Janner et al., 2022; Du et al., 2023; Zhang et al., 2024). However, while the physical world is three-dimensional in nature, existing world models operate in the space of 2D pixels. As a result, existing models fail to provide information about the precise state of world, such as how far away a manipulated object is or what 6-DoF pose is needed to accurately manipulate an object, which is often important to extract precise controls to execute a robot. In addition, simulated dynamics will often not be consistent in the 3D world, such as having objects change dramatically in size and shape over time, making it difficult to use such models for effective data driven simulation and generation.

In this paper, we explore how we can instead learn a 4D embodied world model, which directly simulates the dynamics of 3D world. By directly modeling the dynamics of the world in 3D, we can more accurately simulate physical interactions, such as the grasping of an object or the opening of drawer. Simultaneously, the detailed output space of 3D dynamics allows us to precisely extract detailed 6-DoF actions of a robot directly from the 3D shape of the manipulator, allowing our world model to effectively serve as a zero-shot policy.

However, the task of learning a 4D embodied world model is very challenging as the dynamics of the world is extremely computationally expensive to train and learn, requiring models to generate outputs in three-dimensional space and time. To efficiently represent and predict the dynamics of the world, we propose a substantially more lightweight representation of the 4D world, consisting of predicting a sequence of RGB, depth and normal maps of the scene. This combined representation accurately captures the appearance, geometry, and surface of a scene while being substantially lower

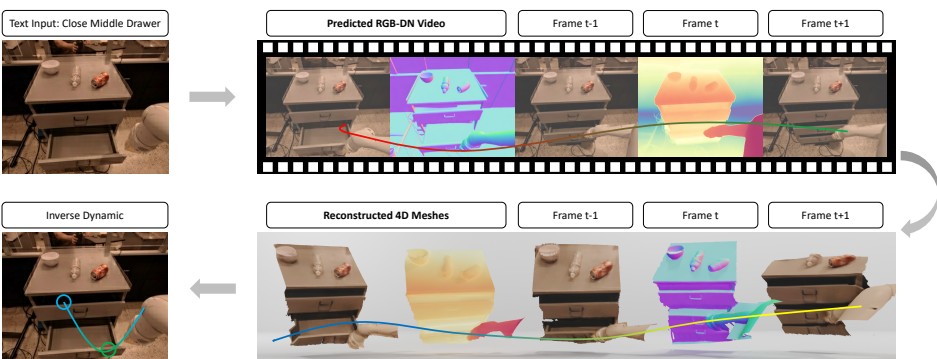

Figure 1: **4D Embodied World Models.** Our approach gets an input image and instructions and predicts RGB-D-N (RGB, Depth, Normal) maps. We propose a normal integration method which constructs a high-quality 4D mesh of the interaction from these predictions.

dimensional than explicitly predicting world dynamics. Furthermore, such a representation shares substantial similarities to existing video space models, allowing us to directly use the generative capabilities and architecture improvements of existing video space models to effectively construct our 4D world model.

Given this intermediate representation, we present an efficient algorithm to reconstruct accurate 4D scenes from generated maps. For each frame, we use a combination of both depth and normal prediction to integrate a smooth 3D surface of scene. We then use optical flow between generated frames to distinguish between background and dynamic regions in the reconstructed 3D scene across frames, and add a loss function to reconstructions to enforce consistency across scenes over time. We find this enables us to construct a fidelity 4D generated meshes for the scene suitable for downstream tasks such as policy prediction for the robot (Figure 1).

A key challenge for training a 4D world model is a lack of access to existing large-scale datasets with existing 4D annotations, or the high-quality image, depth and normal annotations needed to train our approach. To construct such data, in principle, we can use pretrained depth and normal estimators (Ke et al., 2024; Yang et al., 2024) to obtain such estimates from video data, but these estimators are typically limited to predicting relative value maps from individual frames. As a result, as a scene changes, these estimators will produce depth and normal maps with significant inconsistencies across time. To tackle this challenge, we develop a data collection pipeline that leverages optical flow between frames in a video to enforce consistency between generated depth and normal maps across timesteps. In particular, we use optical flow to guide a depth and normal diffusion model across frames in a video, which we find is sufficient to ensure consistent depth and normal predictions across all timesteps without the need for expensive ground-truth annotations, facilitating the training of our world model on a large scale.

Overall, our paper has the following contributions: **(1)** We introduce a 4D embodied world model, and present an efficient representation of this model, in the form of RGB, depth and normal maps, and illustrate how this representation can be used to construct a full 4D mesh of the scene. **(2)** We present a pipeline to automatically extract 4D world model data from existing video datasets, leveraging an optical flow guided depth and normal diffusion model. **(3)** We illustrate the efficacy of the approach in generating consistent 4D meshes across different environments, substantially outperforming other baselines, as well as illustrating its downstream use as an effective policy.

## 2 RELATED WORK

**Embodied Foundation Models** A flurry of recent work has focused on constructing foundation models for general purpose agents (Yang et al., 2023b; Firoozi et al., 2023). One line of work has focused on constructing multimodal language models that operate over images (Li et al., 2022; Jiang et al., 2022; Raman et al., 2022; Driess et al., 2023; Wang et al., 2023; Zhang et al., 2023; Gramopadhye & Szafir, 2022) as well as 3D inputs (Hong et al., 2023; Huang et al., 2023) and output text describing the actions of an agent. Other works have focused on construction of vision-language-

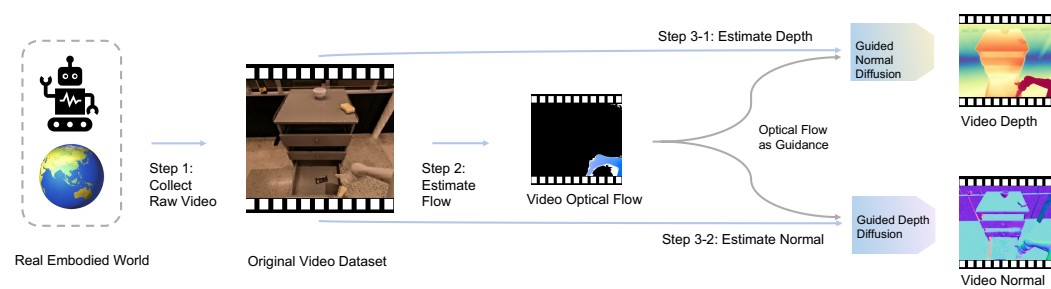

Figure 2: **Data Collection Visualization.** An illustration for the data collection pipeline.

action (VLA) models that directly output action tokens (Brohan et al., 2023; Kim et al., 2024; Zhen et al., 2024). Both of the previous approaches aim to construct foundation model policies (over text or continuous actions). In contrast, our work aims to instead construct a foundation 4D world model for embodied agents, which can then used for downstream applications such as planning (Du et al., 2023; Zhang et al., 2024) or policy synthesis (Du et al., 2024; Liang et al., 2024).

**Learning World Models** Learning dynamics model of the world given control inputs has been long-standing challenge in system identification (Ljung & Glad, 1994), model-based reinforcement learning (Sutton, 1991), and optimal control (Åström & Wittenmark, 1973; Bertsekas, 1995). A large body of work focused on learning world models in the low dimensional state space (Ferns et al., 2004; Achille & Soatto, 2018; Lesort et al., 2018), which while being efficient to learn, is difficult to generalize across many environments. Other works have explored how world models may be learned over pixel-space images (Chiappa et al., 2017; Ha & Schmidhuber, 2018; Hafner et al., 2021; Chen et al., 2022; Micheli et al., 2022), but such models are trained on simple game environments. With advances in generative modeling, a large flurry of recent research has focused on using video models as foundation world models (Yang et al., 2023a; OpenAI, 2023; Zheng et al., 2024; Xiang et al., 2024; Bruce et al., 2024; Zhou et al., 2024) but such models operate over the space of 2D pixels which does not fully simulate the 3D world. Most similar to our work, in Zhen et al. (2024), a world model over 3D inputs is learned. In contrast to this work, our world model directly captures the dynamics of 3D scenes using a compact representation of RGB-DN video.

## 3 4D EMBODIED VIDEO DATA COLLECTION

Learning 4D embodied world models requires large-scale 4D datasets, which are expensive to collect in the real world. In this section, we present a data annotation pipeline that enables us to automatically construct 4D datasets from existing video datasets. For an illustration of this process, see Figure 2. We collected 87,212 episodes of $(\mathcal{V}, \mathcal{T})$ directly from the RT-1 dataset (Brohan et al., 2022), which is a large dataset of real-world tabletop robotic manipulation tasks involving 17 objects.

Given the original input video, $\mathcal{V}$ and a robot text action description $\mathcal{T}$, our data collection protocol aims to automatically obtain 4D annotations of $(\mathcal{F}, \mathcal{D}, \mathcal{N})$ quadruplet structure, where $\mathcal{F}$, $\mathcal{D}$ and $\mathcal{N}$ denotes the optical flow, depth and normal information corresponding to $\mathcal{V}$.

To generate the optical flow $\mathcal{F}$, we employ RAFT (Teed & Deng, 2020), an efficient and high-quality optical flow estimation method suitable for various video datasets. The optical flow between consecutive frames is computed as $\mathcal{F} = \text{RAFT}(\mathcal{V})$. The next step involves generating the 3D annotations: depth $\mathcal{D}$ and surface normal $\mathcal{N}$. These 3D annotations provide richer visual context and enhance the understanding of the 4D environment represented by $\mathcal{V}$. Recent advances in monocular depth and normal estimators, typically based on models like Vision Transformers (ViT) or Diffusion Models, have shown strong generalization capabilities across diverse datasets.

In our approach, we leverage the diffusion model Marigold (Ke et al., 2024) to estimate $\mathcal{D}^i$ and $\mathcal{N}^i$ from each video frame $\mathcal{V}^i$. The frame is first encoded into a latent representation $\mathbf{v}^i$ using a variational autoencoder (VAE) (Kingma, 2013a; Rezende et al., 2014): $\mathbf{v}^i = \text{Encoder}(\mathcal{V}^i)$. At the same time, an initial depth or normal latent map $\mathbf{z}_T^i$ is sampled from a Gaussian distribution: $\mathbf{z}_T^i \sim \mathcal{N}(0, 1)$. Then we iteratively apply the learned denoiser U-Net (Ronneberger et al., 2015) $\epsilon$ to reconstruct the $\mathbf{z}_0^i$ and

$\mathcal{Z}^i$:

$$\mathbf{z}_{t-1}^i = \epsilon(\mathbf{v}^i, \mathbf{z}_t^i, t) \quad \text{and} \quad \mathcal{Z}^i = \text{Decoder}(\mathbf{z}_0^i) \quad \text{where} \quad (\mathbf{z}, \mathcal{Z}) \in \{(\mathbf{d}, \mathcal{D}), (\mathbf{n}, \mathcal{N})\} \tag{1}$$

However, most existing approaches (Yang et al., 2024; Ke et al., 2024; Bhat et al., 2023) are primarily designed for image-based inputs and often struggle when processing videos, resulting in unstable predictions. To overcome this limitation, we propose a novel technique that leverages optical flow as guidance to refine depth estimation. The key insight is that optical flow can capture how static backgrounds and objects move within a scene. Therefore, during the inference stage of the diffusion models, we introduce a loss function that enforces consistency between the backgrounds of consecutive frames. Specifically, we define a static region mask for the $i$-th frame based on the optical flow magnitude: $\mathcal{M}^i = (\|\mathcal{F}^i\| < c)$ where $c$ is a predefined threshold. In practice, we erode the mask to ensure it covers a robust region. Finally, we define the loss function and integrate its gradient into Eq. 1:

$$\mathcal{L}(\mathbf{z}_\mathbf{t}^\mathbf{i}) = \left\| \text{Decoder}\big(\epsilon(\mathbf{v}^i, \mathbf{z}_t^i, t)\big) \circ (\mathcal{M}^i \cap \mathcal{M}^{i-1}) - \mathcal{Z}^{i-1} \circ (\mathcal{M}^i \cap \mathcal{M}^{i-1}) \right\|^2 \tag{2}$$

$$\mathbf{z}_{t-1}^i = \epsilon_\theta \left[ \mathbf{v}^i, \left( \mathbf{z}_t^i - w \boldsymbol{\nabla}_{\mathbf{z}_\mathbf{t}^\mathbf{i}} \mathcal{L} \right), t \right] \tag{3}$$

where $\circ$ represents the element-wise product and $\mathcal{M}_i \cap \mathcal{M}_{i-1}$ selects the overlapping stable regions between two consecutive frames. The pseudocode is presented in appendix Algo. 1.

## 4 LEARNING 4D EMBODIED WORLD MODELS

We introduce the 4D Embodied World Model, which predicts future RGB, depth, and normal maps based on a given input image and text, generating dynamic meshes from these outputs. We leverage a pretrained video diffusion model as the backbone to predict this rich set of 2D geometric information. Then, we propose an efficient method to convert the RGB-DN video into 4D meshes.

### 4.1 PRELIMINARIES ON VIDEO DIFFUSION MODELS

Diffusion models (Ho et al., 2020; Rombach et al., 2022) are capable of learning the data distribution $p(x)$ by progressively adding noise to the data until it resembles a Gaussian distribution through a forward process. During inference, a denoiser $\epsilon$ is trained to recover the data from this noisy state. Latent video diffusion models (Zheng et al., 2024) utilize a Variational Autoencoder (VAE) Kingma (2013a); Van Den Oord et al. (2017), in the latent space of the data, maintaining high-quality outputs while more efficiently modeling the data distribution. In this section, we formulate the task of RGB $\mathcal{V}$, depth $\mathcal{D}$, and normal $\mathcal{N}$ map video generation as a conditional denoising generation task, i.e., we model the distribution $p(\mathbf{v}, \mathbf{d}, \mathbf{n} | \mathbf{v}_0, \mathcal{T})$, where $\mathbf{v}, \mathbf{d}, \mathbf{n}$ represent the predicted future latent sequences of RGB, depth, and normal maps, respectively. The condition $\mathbf{v}_0$ is a given RGB image latent, and $\mathcal{T}$ denotes the instruction provided by the user.

The forward diffusion process adds Gaussian noise to the latent over $T$ timesteps, defined as:

$$q(\mathbf{z}_t | \mathbf{z}_{t-1}) = \mathcal{N}\left(\mathbf{z}_t; \sqrt{\alpha_t} \mathbf{z}_{t-1}, (1 - \alpha_t)\mathbf{I}\right) \quad \text{where} \quad \mathbf{z} \in \{\mathbf{v}, \mathbf{d}, \mathbf{n}\} \tag{4}$$

where $t \in \{1, 2, \ldots, T\}$ denotes the diffusion step, $\alpha_t$ is a parameter controlling the noise influence at each step, and $\mathbf{I}$ is the identity matrix. In the reverse process, the model aims to recover the original latent from the noise. A denoising network $\epsilon_\theta(\mathbf{x}_t, t, \mathbf{v}_0, \mathcal{T})$ with learning parameters $\theta$ is trained to predict the noise added at each timestep. For simplicity, let $\mathbf{x}_t = [\mathbf{v}_t, \mathbf{n}_t, \mathbf{d}_t]$, which denotes the concatenation operation. The reverse process is defined as:

$$p_\theta(\mathbf{x}_{t-1} | \mathbf{x}_t, \mathbf{v}_0, \mathcal{T}) = \mathcal{N}\left(\mathbf{x}_{t-1}; \mu_\theta(\mathbf{x}_t, t, \mathbf{v}_0, \mathcal{T}), \Sigma_\theta(\mathbf{x}_t, t)\right), \tag{5}$$

$$\text{where} \quad \mu_\theta(\mathbf{x}_t, t, \mathbf{v}_0, \mathcal{T}) = \frac{1}{\sqrt{\alpha_t}} \left( \mathbf{x}_t - \frac{1 - \alpha_t}{\sqrt{1 - \bar{\alpha}_t}} \epsilon_\theta(\mathbf{x}_t, t, \mathbf{v}_0, \mathcal{T}) \right). \tag{6}$$

The variance term $\Sigma_\theta(\mathbf{x}_t, t)$ is typically constant. Once the denoised sequence of latent $\mathbf{z}_0$ is obtained, the model reconstructs the final video frames using the decoder network, mapping the latent back to the pixel space: $\mathcal{Z} = \text{Decoder}(\mathbf{z}_0)$.

During training, we randomly select a pair of samples from the dataset $(\mathcal{V}, \mathcal{D}, \mathcal{N}, \mathcal{T})$ and apply Eq.4 to noise the RGB-DN data at timestep $t$, minimizing the following objective:

$$L = \mathbb{E}_{\mathbf{v}_0, \mathcal{T}, t \sim \mathcal{U}(T), \epsilon \sim \mathcal{N}(\mathbf{0}, \mathbf{I})} \left[ \left\| [\epsilon_{\mathbf{v}}, \epsilon_{\mathbf{d}}, \epsilon_{\mathbf{n}}] - \epsilon_\theta(\mathbf{x}_t, t, \mathbf{v}_0, \mathcal{T}) \right\|^2 \right] \tag{7}$$

## 4.2 Video Diffusion Models for Joint RGB, Depth and Normal Predictions

Training a diffusion model to model temporal RGB-DN data is a challenging task. To effectively train RGB video models, large-scale video datasets with billions of high-quality samples are used (Zheng et al., 2024). In contrast, even through automatic annotation, our dataset of RGB-DN data contains only around one million data points, which is insufficient to train a world model from scratch. To address this, we finetune the Open-Sora (Zheng et al., 2024) as our RGB-DN prediction model and directly leverage the pretrained knowledge inside the model to effectively bootstrap our 4D model.

To implement this, we use the temporal VAE Kingma (2013a); Van Den Oord et al. (2017) in Open-Sora (Zheng et al., 2024) to separately encode RGB, depth and normal images for each frame of a video. We then train our video diffusion model to denoise and generate the concatenated RGB, depth and normal images. We directly then expand Open-Sora's input and output channels threefold and finetune the remaining parameters of the model to denoise RGB-DN images. The resultant is then able to directly leverage previous knowledge trained from video data, and use it to predict RGB, depth and normal frames representing the 4D dynamics of a scene.

## 4.3 4D Mesh Reconstruction from RGB-DN Video

After obtaining the RGB-DN video, we further optimize the depth and reconstruct the surface to generate a full final dynamic mesh of the scene. Similar to prior works (Ye et al., 2024; Ke et al., 2024), our depth representation for each image is given by a relative map in the range $[0, 1]$, and thus cannot directly reconstruct the entire scene. While past work has sidestepped this by assuming either a default scale for depth or by directly predicting metric depth, such reconstructions from depth are often coarse and often cause reconstructed planes or walls to be tilted.

We instead leverage the normal maps $\mathcal{N}^i$ and optical flow $\mathcal{F}^i$ between frames in a video to obtain precise depth estimates per pixel from relative depth maps $\mathcal{D}^i$. In particular, we use normal maps to enforce constraints on the surface scene. We then use optical flow between frames to enforce 3D consistency in the scene over time. Both maps in combination with the coarse depth $\mathcal{D}^i$ allow to optimize a refined depth map $\hat{D}$ that corresponds to a consistent 4D scene.

To formalize the process and enforce consistency across frames, we can use the perspective camera model to set constraints on the depth and surface normal. In the coordinate system of the 2D image at frame $i$, a pixel position is given as $\boldsymbol{u} = (u, v)^T \in \mathcal{V}^i$, and its corresponding depth scalar, normal vector is $d \in \mathcal{D}^i, \boldsymbol{n} = (n_x, n_y, n_z) \in \mathcal{N}^i$. Under the assumption of a perspective camera whose focal length is $f$ and the principal point is $(c_u, c_v)^T$, as proposed by (Durou et al., 2009), the log-depth $\tilde{d} = \log(d)$ should satisfy the following equations: $\tilde{n}_z \partial_u \tilde{d} + n_x = 0$ and $\tilde{n}_z \partial_v \tilde{d} + n_y = 0$ where $\tilde{n}_z = n_x(u - c_u) + n_y(v - c_v) + n_z f$. In addition, we can add, assumption that assumes all locations are smooth surfaces (Cao et al., 2022), we can convert the above constraint to the quadratic loss function, allowing us to find the minimized depth map:

$$\min_d \iint_\Omega (\tilde{n}_z \partial_u \tilde{d} + n_x)^2 + (\tilde{n}_z \partial_u \tilde{d} + n_y)^2 \mathrm{d}u \mathrm{d}v. \tag{8}$$

Following Cao et al. (2022), we can convert the above objective to an iteratively optimized loss objective. At iteration step $t$, we can compute the matrix $W(\tilde{d}_t)$ and iteratively optimize for a refined depth prediction $\tilde{d}_{t+1}$:

$$\tilde{d}_{t+1} = \arg\min_{\tilde{d}} (A\tilde{d} - b)^T W(\tilde{d}_t)(A\tilde{d} - b) \overset{\text{def}}{=} \arg\min_{\tilde{\mathcal{D}}} \mathcal{L}(\tilde{\mathcal{D}}, \mathcal{N}^i), \tag{9}$$

where $A$ and $b$ are defined by predicted normals and camera intrinsics.

The above optimization approach optimizes depth frame by frame, which lacks temporal consistency across the dynamic scene. To address this, we compute optical flow between frames (Teed & Deng,

2020) $\mathcal{F} = \text{RAFT}(\mathcal{V})$ and enforce consistency of depth across frames. We define the static regions of each frame as the pixels with the magnitude of optical flow is smaller than threshold $\mathcal{M}^i = \|\mathcal{F}^i\| \le c$. We then define the dynamic parts of an image $M_d^i$ as all regions not in $\mathcal{M}^i$. We further define the background of an image $\mathcal{M}_b^i$ as static regions that are fixed across image frames, $\mathcal{M}_b^i = \mathcal{M}^i \cap \mathcal{M}^{i-1}$

Since optical flow represents the movement of objects in the 2D-pixel space, we can retrieve the depth at any position from the previous frame to impose consistency constraints. To compute the depth values from the previous frame at positions corresponding to the current frame, we utilize the optical flow $\mathcal{F}^{i \to (i-1)}$. For each pixel $(u, v)$ in frame $i$, the optical flow provides the displacement $(\Delta u, \Delta v)$, allowing us to find the corresponding pixel in frame $i - 1$ at position $(u - \Delta u, v - \Delta v)$. Based on this mapping, we define the $\mathcal{D}^{i \to (i-1)}$ such that: $\mathcal{D}^{i \to (i-1)}(u, v) = \mathcal{D}^{i-1}(u - \Delta u, v - \Delta v)$. We then introduce the loss function $\mathcal{L}_d$ for dynamic regions of an image:

$$\mathcal{L}_d(\tilde{\mathcal{D}}, \hat{\mathcal{D}}^{i-1}, \mathcal{M}_d^i, \mathcal{F}^i, \mathcal{F}^{i-1}) = \left\| \tilde{\mathcal{D}}^i \circ \mathcal{M}_d^i - \mathcal{D}^{i \to (i-1)} \circ \mathcal{M}_d^i \right\|^2. \tag{10}$$

In addition to the loss terms $\mathcal{L}$ defined previously, we incorporate regularization loss $\mathcal{L}_g$ enforcing that optimized depths are similar to the generated depth map $\mathcal{D}^i$. We define the regularization loss $\mathcal{L}_g$ as:

$$\mathcal{L}_g(\mathcal{D}_1, \mathcal{D}_2, \mathcal{M}) = \|\mathcal{D}_1 \circ \mathcal{M} - \mathcal{D}_2 \circ \mathcal{M}\|^2 \tag{11}$$

We then define a regularization term on optimized depth maps over background regions of images $\lambda_{g2} \mathcal{L}g(\tilde{\mathcal{D}}, \hat{\mathcal{D}}^{i-1}, \mathcal{M}_b^i)$ enforcing that optimized depths for background regions of an image are consistent between frames and a regularization term over dynamic regions of images $\lambda_{g1} \mathcal{L}_g(\tilde{\mathcal{D}}, \mathcal{D}^i, M_d^i)$ enforcing that optimized depth of dynamic regions of an image match with predicted dynamic depths.

Our overall loss objective we optimize is given by:

$$\arg\min_{\tilde{\mathcal{D}}} \ \mathcal{L}(\tilde{\mathcal{D}}, \mathcal{N}^i) + \lambda_b \mathcal{L}_g(\tilde{\mathcal{D}}, \hat{\mathcal{D}}^{i-1}, \mathcal{M}_b^i) + \lambda_d \mathcal{L}_d(\tilde{\mathcal{D}}, \hat{\mathcal{D}}^{i-1}, \mathcal{M}_d^i, \mathcal{F}^i, \mathcal{F}^{i-1}) + \tag{12}$$

$$\lambda_{g2} \mathcal{L}_g(\tilde{\mathcal{D}}, \mathcal{D}^i, M_b^i) + \lambda_{g1} \mathcal{L}_g(\tilde{\mathcal{D}}, \mathcal{D}^i, M_d^i). \tag{13}$$

We initialize the starting depth $\tilde{d}_0 = \mathcal{D}^i$ with the generated depth map, and similar to prior work (Cao et al., 2022; Xiu et al., 2023), where we repeatedly optimize $\tilde{\mathcal{D}}^t$ across multiple iterations (using the previously optimized depth map $\tilde{\mathcal{D}}^{t-1}$ to define the new optimization objective).

Finally, we construct faces by connecting pixels to their nearby neighbors. For mesh denoising, we remove isolated vertices based on mean neighbor distances and eliminate small clusters using DBSCAN (Hahsler et al., 2019). We also discard faces with abnormal normals or high edge-length variance, ensuring the final mesh is cleaner.

### 4.4 INVERSE DYNAMICS MODELS FROM 4D MESHES

After generating 4D meshes, which encapsulate both spatial and temporal information, we extract geometric details that can significantly enhance downstream tasks in robotics. The detailed geometry captured by these 4D meshes plays a crucial role in robotic grasping tasks.

To achieve this, we employ an inverse dynamics model built on the 4D meshes, predicting the appropriate robot action $a_i$ based on the current state $s_i$ and the predicted future state $s_{i+1}$. Mathematically, this relationship is expressed as $a_i = \text{ID}(s_i, s_{i+1})$. In our scenario, $s_i$ represents the scene at time step $i$. Specifically, we sample the meshes to obtain point clouds, which are encoded by a PointNet (Qi et al., 2017) architecture within the inverse dynamics model to extract features. These features, combined with the instruction text embeddings, are further processed by an MLP to generate the final action.

## 5 EXPERIMENTS

In this section, we evaluate the performance of our proposed model across several tasks. In Section 5.1, we present our experiments on 4D mesh prediction using the RLBench (James et al., 2020) and RT-1 (Brohan et al., 2022) datasets. In Section 5.2, we conduct experiments on embodied novel view

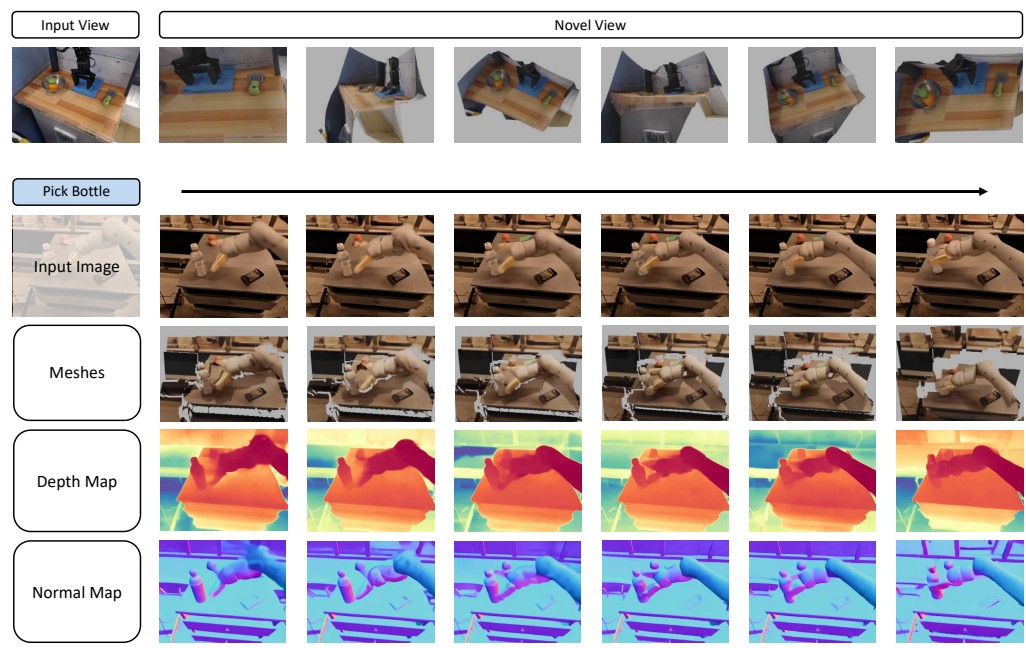

Figure 3: **4D Reconstruction and Video Generation.** The first row shows the results of our 4D reconstruction method from multiple views on a single frame. The following rows present the inference results of our method given the input image and instruction, along with the reconstructed meshes.

| Method | Chamfer $L_1$ | |
| --- | --- | --- |
| | RLBench | RT-1 |
| RGB to depth | 0.2570 | 0.3013 |
| 4D Point Cloud Diffusion | 0.1086 | 0.2211 |
| Ours | **0.0945** | **0.2022** |

Table 1: Comparison for 4D Generation Methods on RLBench and RT-1 Datasets

| Method | Time Cost | Consistency | |
| --- | --- | --- | --- |
| | | Depth | Normal |
| Open-Sora | 25.6 seconds | 0.09267 | 0.04153 |
| Marigold-LCM | **15.2 seconds** | 0.09453 | 0.04647 |
| Guided Marigold | ∼3.5 hours | **0.07299** | **0.03822** |

Table 2: Comparison of Data Generation Methods

synthesis, using RLBench to assess our model's ability to generate novel views from monocular video inputs. In Section 5.3, we explore embodied action planning, applying our model to guide robotic arm policies for specific tasks. Finally, in Section 5.4, we present discussions and ablation studies that analyze the effect of different architectural and data generation choices on the quality and consistency of our video diffusion models.

## 5.1 4D MESH PREDICTION

Since no prior work directly generates dynamic meshes from image and text inputs, we primarily compare our method to a 4D point cloud diffusion model. Our baselines include two main approaches: the first is a 16-frame RGB diffusion model, where we obtain depth from the pretrained depth estimator Marigold (Ke et al., 2024), and lift it to 3D via camera intrinsic and extrinsic parameters. We also modify the Point-E (Nichol et al., 2022) model by conditioning it on the mean of CLIP (Radford et al., 2021) features extracted from both text and image inputs, outputting a point cloud of size $T \times$ num of points, where T is set to 4 due to computational constraints. The datasets used for evaluation include RLBench (James et al., 2020) and our annotated RT-1 (Brohan et al., 2022) dataset. For evaluation, we use the $L_1$ Chamfer Distance metric, which measures the distance between two point sets. The results are shown in Table 1.

As shown in the table, our method achieves the lowest Chamfer distances on both the RLBench (James et al., 2020) and RT-1 (Brohan et al., 2022) datasets, indicating a more accurate reconstruction of 4D structures compared to the baselines. All methods perform better on RLBench (James et al., 2020), which is due to it being synthetic data, with less noise and perfectly accurate depth ground

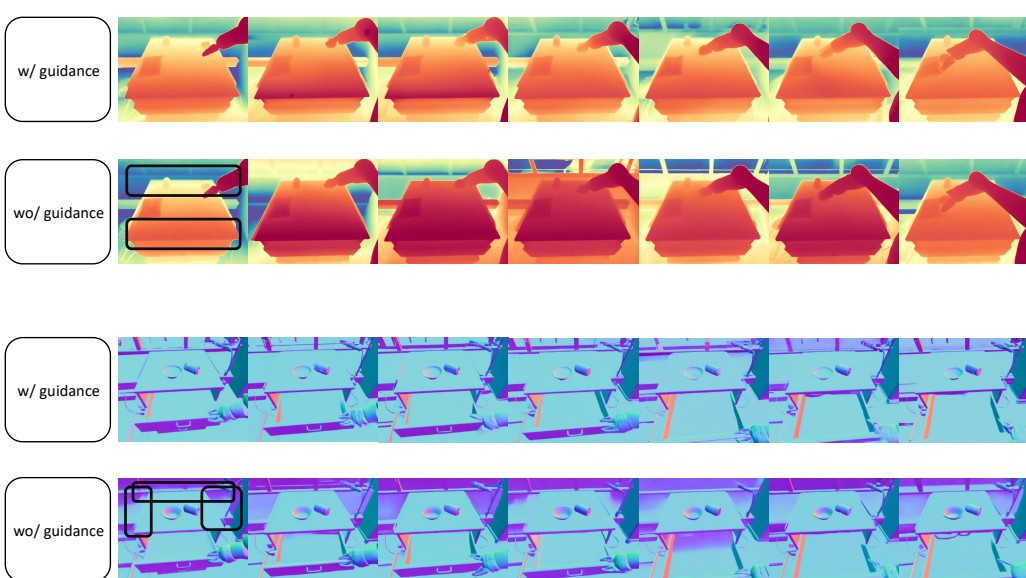

Figure 4: Qualitative Results of Data Generation Methods with and without Guidance, where the black boxes highlight areas of inconsistency.

truth. The RGB-to-depth approach, while simple, suffers from larger errors due to the limitations of depth estimation from 2D images. The 4D point cloud diffusion method performs better, particularly on RLBench, but still lags behind our approach. Additionally, point cloud training is computationally expensive, restricting the number of frames used. In contrast, our model, by leveraging both image and text inputs, manages to generate more precise 3D representations, particularly in capturing fine-grained details in dynamic scenes. We show our qualitative results in Figure 3.

## 5.2 EMBODIED NOVEL VIEW SYNTHESIS

Our method performs monocular video to 4D tasks by predicting depth and normal sequences and generating meshes. We select Shape of Motion (Wang et al., 2024) as our primary baseline, a state-of-the-art video reconstruction approach that utilizes Gaussian splatting (Kerbl et al., 2023). Additionally, we include an RGB-to-depth approach, lifting depth to 3D and rendering point clouds for novel views. Since real-world datasets like RT-1 lack multiview camera information, we conduct experiments on RLBench. The input is a monocular front camera video, and we compare results from the overhead and left shoulder cameras. Metrics include PSNR (reconstruction accuracy), SSIM (structural similarity), LPIPS (perceptual difference), CLIP Score (semantic match) (Zhengwentai, 2023), CLIP aesthetic (visual quality) (LAION-AI, 2022), and Time costs.

| Method | PSNR | SSIM | LPIPS | CLIP Score | CLIP aesthetic | Time costs |
|---|---|---|---|---|---|---|
| Shape of Motion | 10.94 | 0.2402 | **0.7382** | 66.67 | 3.61 | ∼2 hours |
| Ours | **12.99** | **0.4262** | 0.6051 | **83.02** | **3.73** | **∼ 1 minutes** |

Table 3: Performance Comparison of Novel View Synthesis Methods on RLBench Dataset

## 5.3 EMBODIED ACTION PLANNING

Since our world model can predict future scenes, a direct application is to guide robotic arm policies. We compare our method with video diffusion models fine-tuned on OpenSora (Zheng et al., 2024) and use a ResNet (He et al., 2016) in the image-based inverse dynamic model to encode both the current and predicted frames. For simulation, we use RLBench (James et al., 2020) and collect 500 samples for each task to train our model. During inference, given an initial state, we first predict and record all future key frames. In subsequent actions, we only query the inverse dynamic model to

| Methods | Close Box | Close Laptop | Lift Block | Lamp Off |
|---|---|---|---|---|
| Video Diffusion | 81.0 | 14.3 | 19.0 | **57.1** |
| Ours | **95.2** | **28.6** | **23.8** | 42.9 |

Table 4: Evaluation of action planning on RLBench dataset

obtain the corresponding actions by current state and the predicted future state. Table 4 below reports the accuracy across different tasks.

The results show that our method outperforms video diffusion models across the selected tasks. This is because in most tasks, 4D meshes or point clouds can reveal the geometry of objects, providing better spatial guidance for robotics planing, as seen in tasks like Close Box and Close Laptop. However the performance of our model declines in tasks like Lamp Off, due to the small size of the switch, which may not have been sampled. Overall, these results highlight the potential of combining 4D scene prediction with inverse dynamic models to improve robotics task execution.

### 5.4 ABLATIONS

**RGB-DN Video Diffusion Models**. We first conduct ablation studies on our video generation task. We perform experiments to explore the impact of different concatenation methods and the number of frames on the results. For the former, we compare two settings: (1) concatenating RGB-DN images along the width to form a larger image, or (2) using a VAE encoder (Kingma, 2013b; Van Den Oord et al., 2017) to separately process RGB, depth, and normal maps, and concatenating them along the channel dimension before inputting them into the diffusion model. In the latter case, we also modify the input and output dimensions of the backbone network. Our evaluation metrics focus on the generation/reconstruction quality of RGB, depth, and normal maps. Additionally, we introduce an edge similarity metric to assess the consistency across RGB, depth, and normal maps at the same timestamp. Specifically, we convert them into gray-scale images, apply Canny edge detection (Canny, 1986), and compare the edge maps using SSIM (Wang et al., 2004).

| Frames | Channel | RGB | | | Depth | | | Normal | | | Consistency |
|---|---|---|---|---|---|---|---|---|---|---|---|
| | | FVD ↑ | SSIM ↑ | PSNR ↑ | AbsRel ↓ | $\delta_1$ ↑ | $\delta_2$ ↑ | Mean ↓ | Median ↓ | 11.25° ↑ | Edge-sim ↑ |
| 32 | ✗ | 20.12 | 70.23 | 19.32 | 30.22 | 59.21 | 80.28 | 54.22 | 40.87 | 6.41 | 28.27 |
| 32 | ✓ | 19.84 | 69.94 | 19.30 | 18.67 | 69.65 | 89.12 | 19.78 | 10.01 | 26.42 | 31.06 |
| 16 | ✗ | **25.78** | 71.89 | 21.86 | **16.14** | 76.59 | 91.54 | 26.59 | 15.73 | 24.36 | 19.67 |
| 16 | ✓ | 25.45 | **74.98** | **21.94** | 16.53 | **77.13** | **92.15** | **16.23** | **7.78** | **38.50** | **32.98** |

Table 5: **Ablation study.** Impact of the number of frames and channel concatenation. Frames refers to the number of input frames used in the model. "Channel" ✓ indicates concatenating RGB, depth, and normal maps along the channel dimension, while ✗ refers to concatenation along the width.

Although concatenating images along the width results in better RGB reconstruction due to better utilization of the pretrained Open-Sora model (Zheng et al., 2024), it is less effective for depth and normal map predictions. Moreover, the inconsistency between RGB, depth, and normal maps prevents effective post-processing. As shown in Figure 5, concatenating along the channel dimension yields higher-quality depth and normal maps while maintaining consistency across the three value maps. This prevents issues such as a robotic arm appearing in different locations in the RGB and depth/normal maps.

**Data Collection**. This part primarily compares the effects of guidance on depth and normal diffusion models during data generation. As shown in Table 2, "Marigold-LCM" refers to our use of the Marigold Latent Consistency Model (Ke et al., 2024) for independently predicting each frame. "Guided Marigold represents our data generation method. We compare the time cost and the static part $L_1$ difference for these methods. As a reference, we provide the scores from our trained Open-Sora. Qualitative results are presented in Figure 4, where we observe that our proposed data generation method maintains the highest consistency, though at a significantly higher time cost. This also highlights the necessity of training a world model to rapidly predict and generate dynamic scenes.

**Regularization and Consistency Loss** in 4D Mesh Reconstruction. In this task, we evaluate the impact of our newly designed loss terms, as shown in Figure 6. The first two rows demonstrate the

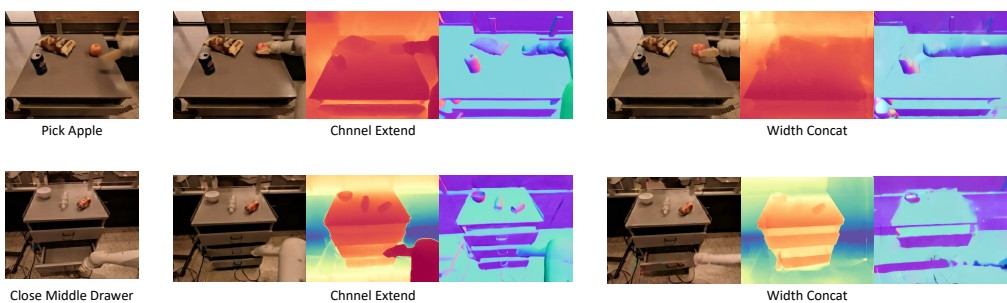

Figure 5: Channel concatenation improves visual quality and ensures consistency across maps.

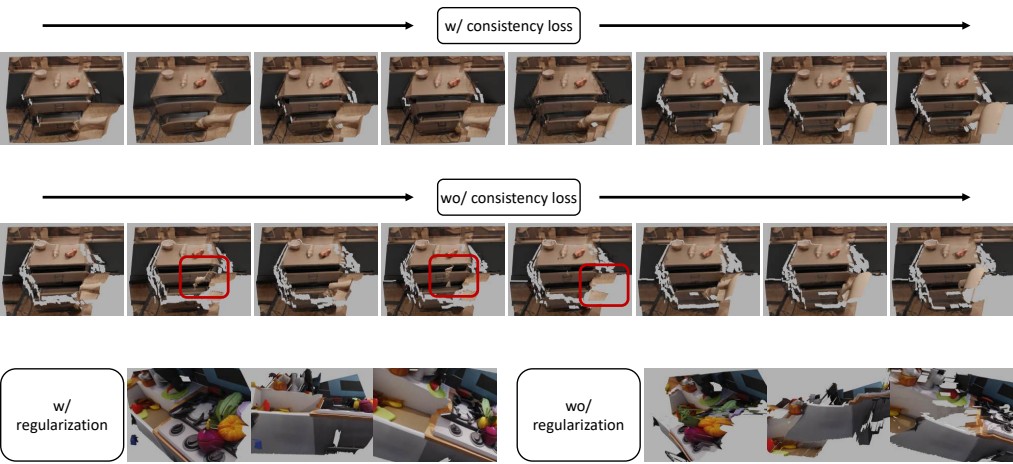

Figure 6: Effect of Consistency and Regularization Losses on 4D Mesh Reconstruction. The red boxes highlight the inconsistent regions.

effect of the consistency loss, where we render frames from the same camera view at different time steps. The results show that the robot arm's movements are more coherent with the consistency loss applied. The last row highlights the role of the regularization loss. We display images of the same frame from three different views, revealing that this loss term helps improve the geometric accuracy of the reconstruction.

# 6 CONCLUSION

Our current approach has several limitations. First, while our RGB-DN representation of a 4D world model is cheap and easy to predict, it only captures a single surface of the world. To construct a more complete 4D world model, it may be interesting in the future to have a generative model that generates multiple RGB-DN views of the world, which can then be integrated to form a more complete 4D world model. In addition, we observe that generated RGB-DN maps from our 4D world model may not be fully consistent with each. Adding additional structure in the architecture or loss function constraints at training time to help enforce consistency is a rich direction of future work.

Overall, our work provides some first steps towards the goal of constructing 4D generative model of the world. We believe that such world models will be increasingly powerful and useful in the future, serving as a way to simulate the physical world and is an important step towards constructing intelligent embodied agents. Such models would then enable us to train policies in the real world in a fully offline manner, as well as roll out and imagine future plans in the world.

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

# A APPENDIX

## A.1 DATA COLLECTION DETAILS

For data generation, we use the Marigold (Ke et al., 2024) DDIM Scheduler, performing 100 denoising steps in total. During the last 30 steps, optical flow guide inference was incorporated. The threshold for background mask computation was set to 0.5, and the gradient descent weight was configured as 2000. The pseudocode for the guidance stage is as follows:

---
**Algorithm 1** Denoising With Optical Flow Consistency

---
1: **Parameters:** $w$                   ▷ Gradient descent weight
2: **Inputs:**
- $\mathbf{z}_t^i$: Noisy depth/normal latent variable at timestep $t$
- $\mathbf{v}^i$: Latent encoding of the video frame $\mathcal{V}^i$
- $\mathcal{M}^i, \mathcal{M}^{i-1}$: Static region masks for the current and previous frames
- $\mathcal{Z}^{i-1}$: Depth/normal annotation from the previous frame
- $t$: Current timestep in the denoising process

3: **Outputs:** $\mathbf{z}_{t-1}^i$: Noisy images at timestep $t-1$
4: **function** DENOISINGONESTEP($\mathbf{z_t^i}$)
5:    $\mathbf{z}_t^i$: require grads ← True
6:    $\mathbf{z} = \epsilon_\theta(\mathbf{v}^i, \mathbf{z}_t^i, t)$               ▷ First denoise
7:    $\mathcal{L} = ||\text{Decoder}(z) \circ (\mathcal{M}^i \cap \mathcal{M}^{i-1}) - \mathcal{Z}^{i-1} \circ (\mathcal{M}^i \cap \mathcal{M}^{i-1})||^2$   ▷ By equation. 2
8:    $\mathcal{L}$.backward()             ▷ To get the value of $\boldsymbol{\nabla}_{\mathbf{z}_t^i}\mathcal{L}$
9:    $\mathbf{z}_t^i \leftarrow \mathbf{z}_t^i - w\boldsymbol{\nabla}_{\mathbf{z}_t^i}\mathcal{L}$
10:    $\mathbf{z}_{t-1}^i = \epsilon_\theta(\mathbf{v}^i, \mathbf{z}_t^i, t)$            ▷ Second denoise
11: **end function**

---

## A.2 VIDEO DIFFUSION IMPLEMENTATION DETAILS

We trained our video diffusion model using the STDiT3-XL/2 architecture, fine-tuned on OpenSora-v1.2 (Zheng et al., 2024), employing $6 \times 8$ V100 GPUs. The model processes videos with 16 frames, using gradient checkpointing to optimize memory usage. For acceleration, we set a batch size of 2, used bf16 precision, and applied the ZeRO2 (Rajbhandari et al., 2020) optimization plugin. Additionally, we leveraged a T5 text encoder (Raffel et al., 2020) for conditioning. For sampling, we use the rflow scheduler with logit-normal sampling across 30 steps and set a classifier-free guidance scale of 7.0.

Training spanned 40,000 iterations with an initial learning rate of 1e-4, 1.0 gradient clipping, and a 1,000-step warmup. The optimizer incorporated Adam with epsilon set to 1e-15, and an exponential moving average (EMA) decay of 0.99 was used to stabilize training.

## A.3 IMPLEMENTATION DETAILS FOR ROBOTICS PLANING TASKS

For the RLBench training, we adopted the same architecture and methods as our video diffusion model, with the primary difference being that we used 13 frames and fine-tuned the model on the RT-1 dataset.

For the action prediction stage, we first filter out the background and floor from the data, focusing only on the points of the table and the objects manipulated by the robotic arm, and then sample 1024 points from filtered the point cloud. In our inverse dynamic model, the PointNet extracts features from this point cloud, which are then concatenated with the instruction's language embedding and passed into a 4-layer MLP, finally outputting the 7DoF actions.

### A.4 4D Meshes Generation

The parameters for the loss term in Eq. 12 are set differently for the RT-1 and RLBench datasets, as shown in the table below:

| Dataset | $\lambda_d$ | $\lambda_b$ | $\lambda_{g1}$ | $\lambda_{g2}$ |
|---------|-------------|-------------|----------------|----------------|
| RT-1    | 20          | 200         | 20             | 20             |
| RLBench | 20          | 200         | 2              | 2              |

Table 6: Loss Term Parameters for RT-1 and RLBench Datasets

