# OpenReview forum: "Learning 4D Embodied World Models"
_ICLR.cc/2025/Conference — ICLR 2025 Conference Withdrawn Submission_

### Official Review · Reviewer_KvjU · 2024-11-02

**Soundness:** 2
**Presentation:** 3
**Contribution:** 3
**Rating:** 5
**Confidence:** 4

**Summary:**

This paper presents a novel framework for constructing 4D world models, leveraging RGB-DN (RGB, Depth, Normal) representations to enhance embodied agents' ability in dynamic, long-horizon tasks.  The approach integrates real-time scene understanding with 4D mesh reconstruction, enabling agents to capture detailed and temporally consistent representations of the environment.  Additionally, this paper introduces a data collection pipeline using optical flow-guided diffusion models for generating consistent depth and normal annotations across frames.  Extensive experiments validate the effectiveness of this model, highlighting its potential for improving multi-step planning and manipulation tasks in realistic settings.

**Strengths:**

1. The framework leverages RGB-DN data to build dynamic 4D representations, allowing embodied agents to capture detailed, temporally consistent scene information, which is essential for tasks requiring real-time scene understanding and long-horizon planning.

2. The authors introduce a data collection approach using optical flow-guided diffusion models, which enables consistent depth and normal annotations across video frames without the need for expensive 4D ground-truth data, making it both practical and cost-effective.

3. The paper provides a thorough evaluation of the model's performance on realistic benchmarks, demonstrating its advantages in long-horizon tasks and its potential for improving embodied agents' capabilities in complex, multi-step scenarios.

**Weaknesses:**

1. For EMBODIED ACTION PLANNING, only RGB information is used as the input of control, while most of the content of this paper is to explain the importance of 3D content. Many current works have proved the feasibility of RGBimage as the input of IDM [1][2][3][4]. The article needs more experiments to prove the advantage of additional information.

2. It is a long pipeline to predict RGB-DN and then build dynamic 4D representations and control the embodied agent through IDM. The real-time performance of the method is worrying.

3. In section 4.4 ai=ID(si,si+1), But immediately after, "These features, combined with the instruction text embeddings, are further processed by an MLP to generate the final action. "The description is inconsistent with the formula

4. In the experimental section, image-based IDM is used by Opensora, which does not accept text input, while the proposed method in question 3 does accept text input. I think it is unfair to compare the effects of the two.

[1]Du, Yilun, et al. "Video language planning." arXiv preprint arXiv:2310.10625 (2023).
[2]Du, Yilun, et al. "Learning universal policies via text-guided video generation." Advances in Neural Information Processing Systems 36 (2024).
[3]Ajay, Anurag, et al. "Compositional foundation models for hierarchical planning." Advances in Neural Information Processing Systems 36 (2024).
[4]Black, Kevin, et al. "Zero-shot robotic manipulation with pretrained image-editing diffusion models." arXiv preprint arXiv:2310.10639 (2023).

**Questions:**

I would like to know more about the experiment and deployment details of EMBODIED ACTION PLANNING, which is a bit brief and vague in this article.

---

### Official Review · Reviewer_UbxB · 2024-11-04

**Soundness:** 3
**Presentation:** 1
**Contribution:** 2
**Rating:** 3
**Confidence:** 4

**Summary:**

The paper proposes a diffusion model that predicts future RGB, depth, and normal images of a scene conditioned on an input image and a language prompt. They further introduced an inverse dynamics model for inferring robot actions from the predicted future. The main purpose of this model is to inform a robot policy for tasks like robot manipulation. The authors trained the model on the RT-1 dataset and evaluated it on the same dataset to compare against several baselines on several tasks.

**Strengths:**

- the problem being studied is an important problem for robot learning and manipulation. It is a very challenging problem.
- the results on 4D prediction seems promising, but how well the model generalizes is questionable.

**Weaknesses:**

I have two major complaints about the paper:
1. the presentation of the paper is poor. The figures are not well-made and sometimes contain obvious errors.
2. the contribution to robot learning is very limited.

- the authors claim that they "collected" the data (e.g. in Fig. 2), but also mentioned that the data is from RT-1. Did the authors collect new data for this work or just reuse data from RT-1? Will the data be publicly released?
- the presentation of this paper is pretty poor. The figures are not very well-made or clear and sometimes contain obvious errors. For example, in figure 2, "guided depth diffusion" and "guided normal diffusion" are flipped. The texts are also not centered or aligned. It is clear that the authors didn't spend enough time polishing the figures which poses some difficulties for readers.
- how generalizable is the model? All of the evaluation seems to be done on in-distribution datasets. What about in-the-wild results? What about other robot form factors like UR5 and Franka?
- one of the main claimed benefits of the proposed model is the benefits to robot policy learning. Is there any actual policy rollouts or robot planning results? If the paper claims contribution to manipulation, evaluation only on prediction quality such as 2d/3d losses is not enough. There's a big gap between a good prediction and a successful robot execution.
- in table 4, why only compare to an invented baseline based on OpenSora? What about typical BC or RL policies? The baseline used doesn't really provide much information since the implementation is not clear.
- is there any deformable objects being tested? From the paper, all figures show only rigid objects and grasping tasks. These tasks don't really require an advanced world model.
- is there any evaluation on the inverse dynamics model?

**Questions:**

Please see weaknesses for questions.

---

### Official Review · Reviewer_4uR4 · 2024-11-04

**Soundness:** 2
**Presentation:** 3
**Contribution:** 3
**Rating:** 6
**Confidence:** 3

**Summary:**

The authors present a 4D world model in the embodied scenarios. In particular, they propose a method for generating dynamic meshes by integrating sequences of 2D observations, including RGB images, depth maps and normal images, utilizing the existing 2D world models like video diffusion models. The methodology encompasses an automated pipeline for annotating large-scale robotic video datasets, coupled with training strategies to ensure 3D consistency. The authors evaluate their approach through multiple tasks, including 4D generation, novel view synthesis, and active planning.

**Strengths:**

1. This work represents the first attempt to construct a world model for 4D meshes, constituting a novel exploration of more informative generative representations for downstream tasks.

2. The authors provide many qualitative and quantitative evidence to show the visual quality of their generated data.

**Weaknesses:**

1. Lack of credit to 3D-VLA. 3D-VLA has proposed the annotation pipeline to estimate optical flow that guide aligned depth estimation for existing videos. However, the paper neither acknowledges 3D-VLA's annotation methodology nor provides a comparative analysis between the proposed method and 3D-VLA.

2. As shown in the website, the consistency of generated RGB-DN flows seem not good enough. This raises concerns regarding the suboptimal quality of dynamic meshes constructed from potentially inaccurate RGB-DN and their subsequent impact on downstream applications.

3. The experimental evaluation would benefit from extended investigation in robotic policy learning. Specifically, the authors could assess the robustness and generalizability of their method across a broader range of tasks in RLBench or alternative benchmarks such as CALVIN.

**Questions:**

1. Given that one of the primary contributions is the automatic annotation pipeline for large-scale video datasets, an analysis of the training data scale's impact would be valuable.

2. While previous works such as 3D-VLA have demonstrated the efficacy of world models for RGB-D and point cloud representations, further justification for the incorporation of normal images and meshes would strengthen the manuscript.

3. A comparative analysis between the proposed method and existing approaches like 3D-VLA in action planning tasks would enhance the evaluation.

4. The proposed consistency and regularization losses warrant quantitative ablation studies to validate their effectiveness.

5. Given that PointNet may not capture local spatial relationships as effectively as more powerful architectures (e.g., PointNet++ or alternative networks), quantitative justification for the choice of point cloud encoder would be beneficial. Moreover, the reported suboptimal performance in Lamp Off tasks, attributed to "the small size of the switch" in Sec. 5.3, may be a consequence of the encoder selection.

6. In my opinion, the term "embodied world model" more precisely refers to an interactive generative model that takes both current state and action as inputs. Therefore, "world model in embodied scenarios" may be a more accurate characterization of the presented work.

7. It seems that the labels "Guided Normal Diffusion" and "Guided Depth Diffusion" in Figure 2 appear to be transposed.

---

### Official Review · Reviewer_TVY9 · 2024-11-08

**Soundness:** 2
**Presentation:** 2
**Contribution:** 2
**Rating:** 3
**Confidence:** 3

**Summary:**

This paper proposes a 4D embodied world model, which, conditioned on image and language instructions, predicts a 4D dynamic mesh that shows how the scene changes over time. The 4D dynamic mesh prediction is achieved by first predicting a sequence of RGB, depth and normal images (while ensuring temporal consistency) and then building a 4D mesh. To learn the world model, existing video robotics dataset are annotated using pretrained depth and normal prediction models. Experiments demonstrate the utility of the world model for 3D prediction, novel view synthesis and embodied action planning.

**Strengths:**

* The idea of predicting 3D structure over time by predicting RGB, depth and surface normals, and integrating them into a 4D mesh is interesting and novel to the best of my knwoledge.
* The idea of generating temporally consistent depth and surface normals from per-frame predictions is also interesting and valuable for automated annotation generation for videos.
* The experiments are well designed to evaluate various aspects of the 4D world model (i.e., mesh prediction accuracy, novel view synthesis and robot action planning). The ablation study in Table 5 is useful to understand the architecture design (i.e., the # of frames inferred and the channelwise concatenation strategy).

**Weaknesses:**

## Approach seems tailor-made for simplistic scenarios
All experiments are performed on simplistic videos in tabletop robot manipulation setting without much camera movements or distractors / clutter. Different aspects of the method rely on the specifics of these settings. For example, using optical flow to infer static parts of the images will work only when the camera is still. This means that both the data annotation generation (i.e., temporally consistent depth and surface normals) and the 4D mesh reconstruction steps are likely to work only in restricted table-top settings. I do not see how such a method can be generalized to robot navigation settings or the even harder egocentric video setting.

## Novelty concerns
The idea of estimating temporally consistent depth is not new. How does the proposed method fare against prior work cited below?

Li, Siyuan, et al. "Enforcing temporal consistency in video depth estimation." _Proceedings of the IEEE/CVF International Conference on Computer Vision_. 2021.
Shao, Jiahao, et al. "Learning Temporally Consistent Video Depth from Video Diffusion Priors." _arXiv preprint arXiv:2406.01493_ (2024).

## Lack of writing clarity
The writing clarity is significantly lacking. This is partly due to the fact that the method builds upon several existing and fairly complex systems. Nevertheless, the approach is hard to completely understand, given the description in the paper. The paper should be self-sufficient to a large degree.

* L174 - 175: Why only one gradient step in Equation 3?
* L216 - 221: "pair of samples" - Not clear what pairs of samples are in this context.
* Equation 7: The noise variables $\epsilon_v, \epsilon_d, \epsilon_n$ are not defined.
* L230 - 236: What is the temporal VAE architecture from Open-Sora? How exactly is the OpenSora architecture modified to accommodate RGB-DN images? Architecture descriptions and diagrams are missing in the paper. Which parameters from the model are finetuned?
* Table 4 - what is the accuracy metric for action planning?
* Section 5.1 - what are the inputs for the 4D mesh prediction task? Is it the entire video or only the first frame?
* Section 5.3: How is the inverse dynamics model trained?

## Missing experiments for "why is 4D embodied world model needed"
A key motivation for the proposed method is that directly modeling the 3D world dynamics can more accurately capture the physical interactions and assist with robotic planning (L043 - 048). However, this core hypothesis has not been sufficiently tested in the paper, i.e., is an embodied 4D world model needed for robotic planning? The only experiment for robotic planning is in Section 5.3 and Table 4 on the RLBench dataset. However, the only baseline selected is a video diffusion model trained on OpenSora. While the 4D world model improves over video diffusion, it has poor accuracies on the majority of the tasks in these simplistic settings. Moreover, relevant methods like Pandora, UniSim, RoboDreamer, 3D-VLA, and This&That (some cited in the related work section) have not been compared against, even though they're useful for action planning. How do these non-4D-world-models fare against the proposed method?

Reference for This&That --- This&That: Language-Gesture Controlled Video Generation for Robot Planning

## Quality-checking depth and surface normals
The annotation approach from Section 3 is interesting, but the quality of annotations has not been sufficiently validated. Table 2 shows preliminary consistency estimates for static regions. However, a more thorough evaluation of the depth and surface normal accuracy is needed on settings like RLBench where ground-truth temporal depth and surface normal estimates can be obtained from the simulator. This is particularly critical since the RT-1 annotations are used for evaluating model predictions in Table 1.

## Other experimental concerns
* Novel view synthesis results look promising in Table 3, but qualitative results are missing to better understand how the difference in metrics is reflected in qualitative visualizations.
* L430 - Has the 4D world model been trained on RLBench data, but the baseline video diffusion model been trained on out-of-domain OpenSora data?

**Questions:**

1. Does the proposed approach scale to more realistic videos (i.e., non-table-top settings)?
2. How does the proposed temporally consistent depth/surface normal prediction compare against prior approaches?
3. The writing clarity needs to be significantly improved. How would the authors address these concerns?
4. Is the 4D embodied world model needed for robot action planning (a key motivating factor for the approach)? Does it outperform non-4d-world models on this task?
5. Can we quantify the depth and surface normals from the temporally consistent estimation approach?

---

### Note · Authors · 2024-11-15

**Comment:**

We appreciate the reviewers' valuable feedback and will address the noted concerns to improve the paper. Thank you for the insightful comments.

**Withdrawal Confirmation:**

I have read and agree with the venue's withdrawal policy on behalf of myself and my co-authors.